# Bioavailability, Speciation, and Crop Responses to Copper, Zinc, and Boron Fertilization in South-Central Saskatchewan Soil

**Noabur Rahman \* and Jeff Schoenau**

Department of Soil Science, University of Saskatchewan, Saskatoon, SK S7N 5A8, Canada; jeff.schoenau@usask.ca
\* Correspondence: mdr422@usask.ca; Tel.: +1-306-881-3170

**Abstract:** An appropriate fertilization strategy is essential for improving micronutrient supply, crop nutrition, yield and quality. Comparative effects of different application strategies of micronutrient fertilizer were evaluated in two contrasting sites/soils (upper slope Chernozem and lower slope Solonetz) within a farm field located in the Brown soil zone of Saskatchewan, Canada. The study objective was to examine the impact of Cu, Zn, and B fertilizer application strategies on their mobility, bioavailability and fate in the soil as well as crop yield responses. The application strategies were broadcast, broadcast and incorporation, seed row banding, and foliar application of Cu, Zn, and B on wheat, pea, and canola, respectively. The study was laid out in a randomized complete block design (RCBD) with four treatment replicates for a specific crop and site. Crop biomass yields were not significantly influenced by micronutrient placement strategies at both sites. Pea tissue Zn concentration (35.2 mg Zn kg$^{-1}$ grain and 5.15 mg Zn kg$^{-1}$ straw) was increased by broadcast and incorporation of Zn sulfate on the Solonetz soil. Residual levels of soil extractable available Cu were increased significantly to 3.18 mg Cu kg$^{-1}$ soil at Chernozem and 2.53 mg Cu kg$^{-1}$ soil Solonetz site with the seed row banding of Cu sulfate. The PRS™-probe supply of Cu (1.84 μm Cu/cm$^2$) and Zn (1.18 μm Zn/cm$^2$) were significantly higher with broadcast application of corresponding micronutrient fertilizer in the Chernozem soil. Both the chemical and spectroscopic speciation revealed that carbonate associated Cu and Zn were dominant species that are likely to control the bioavailability of these micronutrients under field conditions.

**Keywords:** micronutrient deficiency; tissue concentration; bioavailability; fertilizer application strategy; yield response

## 1. Introduction

Reliable prediction of a response to micronutrient fertilization requires an understanding of factors related to soil supply and plant demand. Several factors including the form, rate and placement can all influence micronutrient behaviour and fate in soil and contribute to better nutrition to crops. Although micronutrients are required in small amounts for crop growth, a balanced supply is necessary to achieve to the maximum yield potential as over fertilization can results in plant toxicity [1]. The availability and supply to the plant roots needs to be maintained throughout the crop growth period due to low mobility within the soil–plant system [1,2]. It is generally agreed that the total concentration of micronutrient in soil is not a reliable index of assessing bioavailability and predicting crop responses. Examining the distribution of micronutrients in various soil fractions that differ in their solubility is often used as a useful approach to describing potential micronutrient mobility and bioavailability in soils [3,4]. In most protocols, these fractions are grouped as: (i) water soluble, (ii) exchangeable, (iii) adsorbed, complexed and chelated species, (iv) oxide bound and (v) structurally bound or residual fraction [3,5]. The soil solution and the exchangeable fractions are considered as mobile and bioavailable, and this highly bioavailable portion normally maintains dynamic equilibrium with more slowly available fractions that are adsorbed onto soil mineral and organic surfaces [6]. Therefore,

the fate of applied micronutrient fertilizer, and the resultant bioavailable concentrations in soil solution are closely related to adsorption-desorption, precipitation-dissolution, and complexation mechanisms [6].

Micronutrient transformations in soils are highly dependent on physicochemical properties especially pH, organic matter content, content of free calcium and magnesium carbonates, texture and mineral composition of soil [1,6]. One such example is the high degree of chemisorption of micronutrients such as Cu and Zn in calcareous soils that usually limits their mobility and bioavailability [7,8]. When the pH and the carbonate content of soils are high, Cu and Zn can be occluded as carbonate salts. The subsequent desorption of these micronutrient elements generally requires a high activation energy or a decrease in pH [9]. Thus, the deficiencies of Cu and Zn more likely to occur in calcareous soils under high soil pH. Another mechanism contributing to reduced bioavailability is co-precipitation of these micronutrients as a result of interacting with P in soils that receive continuous high rates of P fertilizer [1,10]. Soil organic matter can also reduce Cu, Zn, and B availability by forming inner-sphere complexes, which are difficult to desorb [11]. However, organic matter can also contribute to enhanced micronutrient availability through the formation of soluble complexes with metal ions. Boron deficiency is observed in coarse textured soils with low organic matter due to high susceptibility of the uncharged species $H_3BO_3$ to removal from the root zone by leaching [12], and potentially low B mineralization. Hence, an understanding of physicochemical factors affecting micronutrient behavior is important when making practical recommendations to growers for best micronutrient fertilizer management.

Micronutrient fertilizer placement has long been recognized an important consideration for optimizing crop uptake and response [13]. Usually, micronutrients are soil-applied or foliar-applied, depending on confirmed deficiency prior to seeding via soil tests or tissue tests as well as visual diagnosis during plant growth in the field [14]. Soil applications are mostly effective in correcting severe deficiencies which are difficult to overcome with a single foliar spray [14]. Additional advantage of soil application is residual benefit over several years [14,15]. However, reduced availability due to physicochemical transformations to more stable forms in the soil may be avoided by using a foliar application method. Furthermore, foliar application of fertilizer-herbicide mixtures may contribute to improving operational efficiencies on farms.

Whether one application method or the other predominates largely depends on crop types, fertilizer products available, soil characteristics that affect degree of anticipated fixation, and initial nutrient levels and anticipated severity of the deficiency. For example, different crops cultivated in rotation vary considerably in susceptibility to micronutrient deficiencies [1,2]. Some crop species are known to release phyto-siderophores (graminaceous plants) and organic acids during deficiency, regulating micronutrient mobility and bioavailability by lowering pH and producing chelation in rhizosphere soils [16]. Moreover, broadcast and incorporation of liquid Cu fertilizer was more effective in increasing wheat yield than granular Cu fertilization in western Canadian soils [15]. The majority of the micronutrient research conducted in Canadian prairies has consisted of agronomic field scale fertilization response studies, with soil assessment limited to the determination of extractable amounts of micronutrients [17–23]. In these studies, crop yield response is emphasized and micronutrient behaviour in soil including fate, mobility and bioavailability is not extensively explored. The study was conducted to evaluate how mobility and bioavailability of different micronutrients is influenced by different application strategies including placement and form, along with crop yield and uptake, using a field research site located in a farm field in south central Saskatchewan.

## 2. Materials and Methods

### 2.1. Field Study Location and Site Description

The field study was located in a 160-acre field at legal land location of SE32- 21- 04-W3 near the town of Central Butte (50°49′40″ N latitude, 106°30′81″ W longitude) in south-

central Saskatchewan, Canada. The soil in this field is classified as Haverhill-Echo complex association, with Brown Chernozem soils intermixed with Brown Solodized Solonetz soils. Brown Chernozems dominate in upslopes while the Solonetz soils are generally found in toe or lower slope positions of catenas. The land topography of the study area is level to gently undulating, and the soil is loamy in texture. These are calcareous soils developed from glacial till parent materials that overlie the outcropping shale or glacial drifts of the Pierre formation [24]. The field was farmed from 1930 to 2011 in a wheat-fallow rotation with no macronutrient or micronutrient fertilizer applied. Since 2012, the field has been in a no-till continuous crop rotation with canola grown in 2013 and wheat grown in 2014, with recommended rates of macronutrients (N, P and S) applied. No micronutrients were applied at any time in the field's history.

Two separate sites within the field, selected to represent higher elevation Chernozem (Haverhill association) and lower elevation (Echo association) soils in the landscape, were used in the study. The spatial variability of soil properties across the landscape usually has a significant control on crop productivity in Prairie agroecosystems [25–27] and is a significant factor of consideration when developing management zones and prescriptions for precision variable rate fertilizer application. On high elevation knolls, nutrient and organic carbon rich topsoil has been moved downwards over the years of cultivation by tillage operations, along with wind and water, which often results in exposure of subsurface inorganic carbonates. Conversely, accumulation of eroded material at lower elevations usually improves fertility and agricultural production potentials at these locations in the field. Therefore, variation in soil type and fertility gradient within a landscape is properly considered in this study and may be used to help understand how response to micronutrient fertilization can change within variable fields of the southern Canadian prairies. Prior to experimental setup, representative soil samples were collected from across the two landscape positions for basic characterization of study sites (Table 1).

**Table 1.** Summary of spring 2015 baseline soil properties (0–15 cm depth) from Haverhill and Echo site locations.

| Soil Association | Basic Properties ♣ | | | | | Extractable Micronutrient (mg kg$^{-1}$) | | |
|---|---|---|---|---|---|---|---|---|
| | pH | EC | FC | OC | Sand | Cu | Zn | B |
| Haverhill | 7.5 | 0.19 | 27.6 | 1.64 | 47 | 0.66 | 0.63 | 0.98 |
| Echo | 7.2 | 0.17 | 26.3 | 1.75 | 42 | 0.73 | 0.71 | 0.92 |

♣ EC = Electrical conductivity (mS/cm); FC = Moisture content at field capacity (%); OC = Organic carbon (%); Sand = Sand (%).

### 2.2. Experimental Setup and Treatments

The experiment was conducted to evaluate the effect of Cu on wheat, Zinc on Pea, and B on canola production at two different slope positions. Experimental set up followed the standard randomized complete block design with four replicates of each treatment. The experiment was laid out in April of 2015, with a total area of 19.5 m $\times$ 31 m for each site, including a 2.5 m pathway between blocks and 8 m distance between crops. The individual experimental plot size is 1 m $\times$ 3 m. For each site (landscape position), the plot size and experimental area were kept small to try to confine the experiment within a relatively uniform area to reduce variability arising from spatial variations in soil properties and micronutrient availability across the study area. Crops were seeded in three rows in each experimental plot. Hard red spring wheat (*Triticum aestivum* var. AC$^{®}$ Waskada), yellow pea (*Pisum sativa* var. Meadow) and canola (*Brassica napus* var. Liberty Link 150) were seeded at the rate of 100, 180 and 5 kg ha$^{-1}$ to provide a desired plant density of 250, 75, and 100 plants m$^{-2}$, respectively. Basal fertilization included N-P-K-S applied just before seeding at the rate of 100-8.7-36.5-17 kg ha$^{-1}$ using the product urea (46-0-0-0), monoammonium phosphate (11-52-0-0) and potassium sulfate (0-0-44-17), respectively. Urea was excluded from the pea plots and the pea were inoculated with

commercial *R. leguminosarium* inoculant (CellTechTM peat based). All of the fertilizers were broadcasted and incorporated with a cultivator to a depth of 4 cm with single pass prior to the micronutrient treatment applications to the soils.

Treatment evaluation in this study involved different application methods of micronutrients including $T_1$: control; $T_2$: soil broadcast; $T_3$: soil broadcast and incorporation; $T_4$: soil seed row banding; and $T_5$: foliar application of Cu on wheat, Zn on pea, and B on canola production. Soil application rate of Cu and Zn was 2 kg Cu, Zn ha$^{-1}$ using sulfate salts (recommended rate), while boric acid was soil applied at a rate of 1 kg B ha$^{-1}$ (recommended rate). Foliar application was made at the rate of 0.25 kg ha$^{-1}$ as chelated Cu, Zn, and boric acid, respectively. All micronutrients were applied as salt, chelate or acid dissolved in water. The solution form with a large water volume was selected to enable uniform application of the micronutrient across the plot area, providing good distribution to reduce variability compared to small amounts of individual granules. A hand sprayer was calibrated to spray at an optimum pressure (30 psi) that maintained uniform solution application to the soils and maximum interception by crop foliage for foliar applied micronutrient.

Soil application of micronutrients ($T_2$, $T_3$, and $T_4$) was performed after basal macronutrient application and incorporation of the macronutrients, and immediately prior to seeding. Broadcast micronutrient consisted of the application of the micronutrient solution across the surface of the plot. Broadcast and incorporation treatment included the incorporation using a cultivator after the application of the micronutrient. For seed-row placement, a hoe opener with 2 cm spread was first manually used to make bands of two to three-centimeter (2–3 cm) depth for seed row banding treatment ($T_4$), deepest for pea, shallowest for canola. After application of the solution micronutrient fertilizer into the furrow, the furrows were covered with soil immediately after micronutrient application, and seed sowing performed on the rows using a single row seeder. This produced a small amount of separation with soil between the seed and the micronutrient fertilizer in the row, to eliminate possible toxicity from direct contact of seed and micronutrient fertilizer. For foliar fertilizer treatments, the products were dissolved separately into 2 L of deionized water and sprayed at vegetative growth stages that would coincide with a companion crop protection application of a herbicide or fungicide and also provide sufficient canopy for foliar interception (tillering stage of wheat and pre-flowering stage of pea and canola, respectively). All other cultural practices including herbicide application were followed according to the requirements and standard recommendations for the study area and crops.

### 2.3. Measurements, Sample Collection and Processing

Representative crop samples were harvested at maturity by taking a 2 m row-length of the middle row in each treatment plot. Collected samples were dried at 40 °C for 7 days in a drying room facilitated by operation of a hot-air circulation system, and after drying the crop samples were threshed mechanically using a rubber belt threshing machine to avoid any source of metal contamination. Straw and grain biomass values were recorded for analyses of yield effects. A random sub-sample of threshed grain and straw materials was then taken and ground using a stainless-steel (chromium, nickel, iron alloy) grinder. Ground plant materials were stored in plastic vials for further chemical digestion in the laboratory to measure total nutrient concentration and uptake for each treatment.

Post-harvest surface soil samples (0–10 cm) were collected and processed by air drying at 25 °C, grinding using a wooden rolling pin and passing through a 2 mm stainless steel sieve. Soil extraction was performed to measure plant available micronutrients and their quantitative distribution in different chemical fractions, in order to assess distribution of soil applied micronutrient fertilizer in the different fractions. Measurements of supply rates were made in-field using PRS$^{TM}$ probes following standard method [28]. In addition to sequential extraction, selected treatment soil samples were scanned using synchrotron facilities at the Canadian Light Source, University of Saskatchewan. for the assessment of chemical forms of Cu, Zn, and B. This is the first work to investigate micronutrient fertilizer reaction products and speciation in Canadian soils. Together, the assessments of plant

availability, nutrient supply rate, chemical and spectroscopic speciation are used to help reveal the fate of the soil applied micronutrient fertilizers in this study.

### 2.4. Extraction Procedures and Analyses

Soil pH and electrical conductivity (EC) were measured in 1:2 soil to water ratio extract [29,30] using a Beckman 50 pH Meter (Beckman Coulter, Fullerton, CA, USA) and an Accumet AP85 pH/EC meter (Accumet, Hudson, MA, USA), respectively. Soil organic carbon (OC) and total carbon (TC) were measured using a LECO-C632 carbon analyzer (LECO© Corporation, St. Joseph, MI, USA) [31]. The pipette method was used to determine soil texture [32], while gravimetric method was used for moisture measurement at field capacity [33].

Extractable available Cu and Zn were extracted using 0.005 $M$ diethylene-triamine-pent-acetic acid (DTPA) extraction [34], and B by hot water extraction [35], respectively. Soil and plant samples were digested with $HNO_3$ + $H_2O_2$ and HCl following the method 3050 A of United States Environmental Protection Agency [36] for total concentration of micronutrients. Along with plant available and total nutrient in soil, a sequential extraction was performed for chemical speciation of Cu, Zn, and B. A modified BCR (Community Bureau of Reference) procedure was used for Cu and Zn extraction [37], while B extraction followed the modified procedure [35] of sequential extractions. Residual fraction is calculated by subtracting all these fractions from the total concentration. Outline of the sequential extraction procedure is provided in Table 2. To provide an assessment of the supply rate of available micronutrient in the field under field conditions over the growing season, Plant Root Simulator (PRS)™ probes (Western Ag Innovations Inc.; Saskatoon, SK, Canada) were installed in each treatment plot to measure soil supply of micronutrients over two-week intervals up to 7 weeks after planting. The working procedure followed for anion-exchange PRS™-probe use, which includes charging, washing and elution, is described in a previous study [38].

**Table 2.** Summary of the sequential extraction methods used for Cu, Zn, and B.

| Extraction Step | Cu and Zn | | B | |
|---|---|---|---|---|
| | Reagent | Nominal Target Phase (s) | Reagent | Nominal Target Phase (s) |
| $F_1$ | 0.11 $M$ $CH_3COOH$ | Solution, carbonate, exchangeable fraction | 0.05 $M$ $KH_2PO_4$ | Specifically adsorbed |
| $F_2$ | 0.5 $M$ $NH_2OH.HCL$ | iron/manganese oxyhydroxide fraction | 0.2 $M$ acidic $NH_4$-oxalate (pH = 3) | Oxide bound |
| $F_3$ | $H_2O_2$ (8.8 $M$) + $CH_3COONH_4$ (1.0 $M$) | organic matter and sulphides bound fraction | 0.02 $M$ $HNO_3$ + 30% $H_2O_2$ | Organically bound |

Solution obtained from extractions and digestions were analyzed for Cu and Zn concentration using a flame atomic absorption spectrophotometer (Varian Spectra 220 Atomic Absorption Spectrometer; Varian Inc.; Palo Alto, CA, USA), while B measurement was performed by 4100 MP-AES (Microwave Plasma-Atomic Emission Spectrometer (Agilent Technologies)). Accuracy of the extraction and digestion procedure for total micronutrient concentrations was verified by comparing with several reference materials such as BCR-701, SRM-1515, SRM-1570a, SRM-1573a and SRM 2709a.

### 2.5. Spectroscopic Analysis and Data Processing

The X-ray absorption spectroscopy (XAS) analyses were performed for solid state molecular level species identification of Cu, Zn, and B using the synchrotron radiation facilities of Canadian Light Source (CLS), Saskatoon. The K-edge XANES spectra was collected at the Hard X-ray MicroAnalysis (HXMA) beamline (06ID-1) at a specific energy range of 8950–9050 eV and 9600–9750 eV for Cu and Zn, respectively. Standard metal foil was used to calibrate the absorption edge energy (8979 eV for Cu and 9659 eV for Zn,

respectively) prior to data collection. The Cu and Zn K-edge spectra were collected in fluorescence mode using a 32 element Ge detector. The B K-edge spectra measurement was performed in the 180–220 eV region using the Variable Line Spacing Planar Grating Monochromator (VLS PGM) beamline (11ID-2). Both the total electron yield (TEY) and the total fluorescence yield (TFY) measurement modes were employed for B K-edge spectra collection in absorption chamber. Soil samples and reference mineral materials were ground with a mortar and pestle and mounted on sample holders as dry powder using KAPTON tape (Kapton® polyimide film) and carbon tape for scanning. The collected XANES spectra were processed using the ATHENA software, Demeter 0.9.23 [39] following the standard procedures of background removal and data normalization.

*2.6. Statistical Analysis*

Statistical analyses were performed separately for each crop and slope position using PROC MIXED procedure of SAS 9.4 [40]. Prior to ANOVA, testing of the assumption of normal distributions (PROC UNIVARIATE) and homogeneity of variances (Bartlett's test) of all data sets were verified, and if needed data transformation was performed following a standard method. Mean separation for significant effects were performed using Tukey's honestly significant difference test at 5% probability levels, while groupings were done using the pdmix800 SAS macro [41].

## 3. Results and Discussion

*3.1. Crop Yield Response to Micronutrient Placement Strategies*

The grain and straw yields of wheat, pea, and canola were not increased by any of the micronutrient application strategies at the two sites (Haverhill soil association and Echo soil association) in the farm field near Central Butte in 2015 (Figure 1). It is important to note that total rainfall at this site during May and June 2015 was only 9 mm versus the historical (30 years) long-term average of 116 mm (https://www.meteoblue.com/ (accessed on 15 May 2018). Furthermore, pre-seeding soil test results indicated that extractable plant available micronutrient concentrations at both were in the marginal to sufficient range. For example, soil DTPA-extractable Cu levels of both Haverhill and Echo site soils in the 0–15 cm depth were above the critical concentration of 0.4 mg Cu kg$^{-1}$ soil (Table 3; [21]). Usually, the crop yield responses to micronutrient fertilization are often influenced by local soil and environmental factors. Considerable research has confirmed the potential incidence of Cu deficiency, and significant yield response of wheat to Cu fertilization on a broad range of western Canadian soils [15,18,21,23,42,43]. Broadcast and incorporation of Cu sulfate was often effective in correcting the deficiency problem to optimize wheat yield [21,43]. To our knowledge, there is no documented evidence of profitable wheat yield response of Cu fertilization on marginally deficient soils with a DTPA-extractable Cu concentration range of 0.4 to 1.0 mg Cu kg$^{-1}$ [21,43]. In this study there was also no response of wheat to Cu addition on "marginally deficient" soils. Therefore, initial Cu fertility assessment should be taken into consideration prior to making a fertilization decision. There was also no yield differences between the Haverhill and Echo association sites. (Figure 1). This suggests that little would be gained from managing the different soils and landscape positions differently in a Cu fertilization plan for the field.

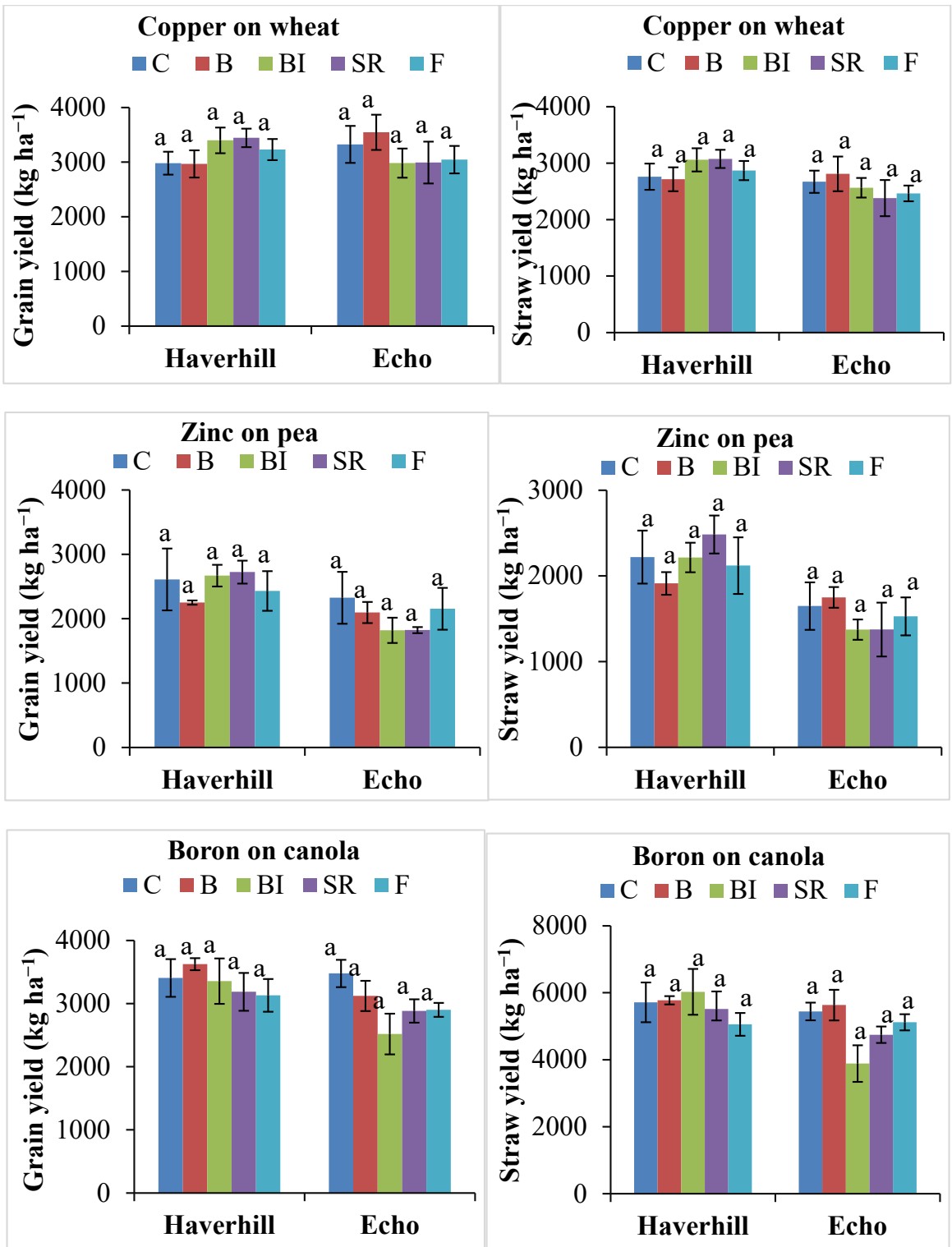

**Figure 1.** Effect of different placement methods of Cu, Zn, and B fertilizers on biomass yield of wheat, pea and canola, respectively at Central Butte field location in 2015. Crops were grown at two sites in the field representing two different landscape elevation positions classified as Haverhill (upslope) and Echo (downslope) soil associations, respectively. Treatment evaluation includes C: control; B: broadcast; BI: broadcast and incorporation; SR: seed row banding; and F: foliar methods of fertilizer application. Treatment columns for a site followed by the same letter are not significantly different ($p > 0.05$). Error bar represents standard error of mean.

**Table 3.** The concentration of Cu, Zn, and B in grain and straw of wheat, pea, and canola, respectively, as affected by different fertilizer application strategies.

| Treatment | Cu in Wheat (mg Cu kg$^{-1}$) | | Zn in Pea (mg Zn kg$^{-1}$) | | B in Canola (mg B kg$^{-1}$) | |
|---|---|---|---|---|---|---|
| | Grain | Straw | Grain | Straw | Grain | Straw |
| | | | Haverhill | | | |
| C | 6.96 a | 2.56 a | 35.1 a | 6.29 a | 10.2 a | 17.7 a |
| B | 8.68 a | 3.77 a | 36.4 a | 6.95 a | 10.5 a | 18.4 a |
| BI | 7.83 a | 4.88 a | 37.1 a | 9.14 a | 10.6 a | 19.2 a |
| SR | 6.91 a | 3.91 a | 37.6 a | 7.48 a | 10.2 a | 17.9 a |
| F | 7.14 a | 3.73 a | 35.5 a | 6.43 a | 10.4 a | 18.2 a |
| *p*-values | 0.602 | 0.359 | 0.658 | 0.723 | 0.602 | 0.412 |
| SEM | 0.900 | 0.771 | 1.54 | 1.59 | 0.268 | 0.571 |
| | | | Echo | | | |
| C | 7.36 a | 4.34 a | 31.1 b | 3.72 b | 11.0 a | 18.8 a |
| B | 7.24 a | 4.85 a | 33.0 ab | 4.83 ab | 10.7 a | 18.1 a |
| BI | 7.54 a | 4.88 a | 35.2 a | 5.15 a | 10.9 a | 19.7 a |
| SR | 7.55 a | 4.91 a | 31.3 b | 4.35 ab | 10.9 a | 19.2 a |
| F | 7.49 a | 3.96 a | 31.3 b | 4.31 ab | 11.4 a | 18.7 a |
| *p*-values | 0.980 | 0.909 | 0.012 | 0.034 | 0.390 | 0.270 |
| SEM | 0.429 | 0.859 | 0.833 | 0.294 | 0.201 | 0.656 |

Treatment evaluation includes C: control; B: broadcast; BI: broadcast and incorporation; SR: seed row banding; and F: foliar methods of fertilizer application. Cu, Zn, and B fertilizer was applied for growing wheat, pea, and canola, respectively. Crops were grown at two sites in the field representing two different landscape elevation positions classified as Haverhill (upslope) and Echo (downslope) soil associations, respectively. Means followed by the same letter in a column for treatments at the same site are not statistically significant (*p* > 0.05). SEM = Standard error of mean (*n* = 4).

Pea yield did not respond significantly to the addition of Zn (Figure 1), likely to be related to both the initial Zn status of the soil and the dry growth environment experienced in 2015. Both trial sites were marginal in available Zn according to soil test (Table 3) and in the zone where pulse response to Zn application is variable and infrequent (0.6 to 0.7 ppm) [20,44,45]. For example, lentil did not respond to Zn fertilization on a marginally deficient (DTPA-extractable Zn > 0.5 mg kg$^{-1}$; [46]) field site in Saskatchewan [45]. Conversely, significant yield response of dry bean was recorded to broadcast and incorporation of 5 kg Zn ha$^{-1}$ in different field sites of Manitoba containing less than 0.5 mg kg$^{-1}$ DTPA-extractable Zn [46]. They also found that foliar application of Zn was less effective than soil application. Additionally, soil applied Zn sulfate at the rate of 5 kg Zn ha$^{-1}$ was effective in increasing grain yield of lentil on two out of ten Saskatchewan soils [44]. However, the yield response was inconsistent as eight of the ten soils were identified as Zn deficient according to soil test [44]. Generally, these contrasting Zn responses are soil based and often influenced by different soil parameters or crop genotypic influence that alter Zn solubility in rhizosphere [44,47,48]. Indeed, the most consistent efficacy of Zn fertilization in optimizing pulse yield is in soils with DTPA-extractable Zn concentrations of less than 0.5 mg Zn kg$^{-1}$ [44,46,49]. Very dry spring conditions may also have limited response to added micronutrient fertilizer in the current study.

The lack of yield response to B fertilization was reflected in no significant effect of different B application strategies on canola (Figure 1). Both experimental sites were identified as sufficient in available B (HWSB~1 mg kg$^{-1}$; Table 3), so a significant response was not expected. Earlier research [22,50,51] reported that B application was effective in increasing seed yield of canola mostly on deficient soils confirmed with a hot water soluble B level of less than 0.5 mg kg$^{-1}$ soil. However, the responses are not sufficiently consistent to warrant widespread application of B. No significant yield responses to B were also observed in many instances, some even in soils rated as highly B deficient according

to soil test [22,51,52]. This inconsistent response may be linked to the strong influence of soil environments on B availability to plants. Reduced availability and supply of B are sometimes associated with dry soil conditions [53,54]. However, dry conditions in 2015 in the current study did not appear to contribute to any deficiency. Additionally, some soils exhibit B toxicity under dry conditions, particularly in arid and semi-arid areas where soils and irrigation water contain high levels of B [55]. In this study, low soil moisture that limited yield potential and B demand and good inherent B fertility of the experimental sites were likely to be the most influential factors associated with lack of canola yield response to B fertilization.

It is also noteworthy that yield depression with Cu, Zn, and B fertilization was observed at the Echo association site, indicating some antagonistic effects under the studied soil-climatic conditions. These results agree well with the study that reported similar yield reduction of canola to B fertilization at several field sites in western Canada [22]. Another study reported that increasing available B levels in soils up to 2 mg B kg$^{-1}$ can induce toxicity and resulted in decreased yield of bean [53]. For Zn fertilization, there was up to 45% grain yield reduction of small red lentil with the addition of 5 kg Zn ha$^{-1}$ on Melfort soil (DTPA-extractable Zn = 1.2 kg ha$^{-1}$) [44]. There is an apparent soil-specific effect on negative yield responses to micronutrient in prairie soils. However, there is no convincing evidence that fertilization increased micronutrient concentration in soils and crops to toxic levels. Still, unnecessary application of micronutrient should be avoided as it can result in substantial yield reductions.

*3.2. Tissue Concentration*

Mean tissue concentrations of Cu and B did not increase significantly from micronutrient fertilizer addition in wheat and canola, respectively, regardless of method of application (Table 3). The average Cu concentrations in wheat grain and straw were 7.0 and 2.6 mg kg$^{-1}$, respectively, in unfertilized control treatment (Table 3). These results suggest that Cu nutrition was in the sufficient range without fertilizer supplement. It was reported earlier that the Cu concentration 2.3 mg kg$^{-1}$ in the grain and 3.9 mg kg$^{-1}$ in the straw of wheat were adequate to produce optimum yield [56]. In young leaf tissue, a concentration of less than 1.5 mg Cu kg$^{-1}$ was considered as critically deficient, and at that point wheat would likely respond to Cu fertilization [57]. Similarly, canola exhibits B deficiency symptoms with youngest leaf tissue concentration of less than 15 mg B kg$^{-1}$ [50]. It was also reported that youngest leaf tissue concentration of 10 to 14 mg B kg$^{-1}$ was associated with yield reduction in canola [58]. Additionally, the B concentration in mature leaves is not considered as a reliable indicator for measuring B requirement in canola [58]. The whole plant tissue concentration range of 20 to 30 mg B kg$^{-1}$ at flowering has been deemed sufficient for normal growth and seed set [59]. Boron levels in mature straw of around 18 mg B kg$^{-1}$ suggest sufficiency of canola based on results of the current study. Overall, micronutrient concentrations in plant tissues could be used to diagnose nutritional deficiency and need for additional fertilizer for crops.

The critical Zn concentration in pea grain at maturity is reported to be less than 20 mg kg$^{-1}$ [60]. In the current study, Zn concentration in the pea grain ranged from ~31 to 38 mg kg$^{-1}$, consistent with Zn sufficiency and the observed lack of yield response to Zn fertilization. Broadcast and incorporation of Zn sulfate on Echo soil significantly increased the Zn concentration in both grain and straw of pea (Table 3). Earlier research [61,62] reported that Zn fertilization was effective in improving yield and quality of pea grown in Zn deficient soils. Further, Zn fertilization is considered as an effective approach of agronomic biofortification to enrich grain Zn concentration and nutritional quality improvement for human consumption [63]. For instance, the bioavailable Zn concentration in pea seeds was increased by increasing the Zn supply to the plants [64]. However, Zn fertilization is often effective only on weathered tropical and sub-tropical soils where both soil and foliar applications of Zn were effective in improving Zn status in edible grains. Lack of any large response of Zn concentration in pea to Zn fertilization is consistent with Zn availability in

the soils and also reflected from the concentrations of Zn in the grain and straw that are above critical levels in the unfertilized control.

### 3.3. Mobility and Bioavailability of Micronutrients in Soils

Sound micronutrient management generally requires some type of reliable assessment of nutrient supplying capacity of the soil to determine the fertilizer needs for a specific cropping regime. Soil testing not only helps to determine the pre-seeding micronutrient requirement but also the potential residual benefits of applied fertilizer in crop rotation [65]. There is considerable research evidence for soil placement of micronutrients providing residual benefits to the following crops over several years [14,43,66]. In this study, higher concentrations of DTPA extractable Cu (0–15 cm) in post-harvest soils was found in the seed row banding of Cu sulfate treatment in both Haverhill and Echo trial sites. Increased Cu concentration in the localized placement site of the seed-row agrees with the restricted mobility of Cu in soils [65,67]. It was also observed that Cu did not move through a soil column even up to 1 cm distance in most of the western Canadian soils evaluated [68]. Micronutrient uptake occurs later in the season when roots have extended further away from the seed-row. On the contrary, broadcasting and incorporation facilitates more even distribution of nutrients throughout the root zone which could have resulted in greater uptake later in the season and also increased fixation of applied Cu due to greater contact with soil. In contrast, the amount of DTPA extractable Cu in the broadcast and incorporation treatment was higher than the seed row placement when Cu sulfate was applied at the rate of 4 kg Cu ha$^{-1}$ up to 4 consecutive years [66]. As expected, foliar application resulted in the lowest residual soil micronutrients of the application strategies, explained by the low rate of application (0.25 kg Cu ha$^{-1}$) in the foliar application versus the soil applied treatments (2 kg Cu ha$^{-1}$).

Surface broadcast of Zn sulfate resulted in significantly higher DTPA extractable Zn in the 0–15 cm soils of Haverhill site, while similar increment was found in the broadcast and incorporation treatment at Echo site (Table 4). Some of this surface broadcast Zn may have been too far away from the roots to be accessible over the season, especially under the dry conditions of the study. Broadcast and incorporation of $ZnSO_4$ at 10 kg Zn ha$^{-1}$ significantly increased the DTPA extractable Zn level in critical to marginally deficient soils of Saskatchewan in early work in the 1980s [20]. It was found that a single broadcast and incorporation of 8 l b Zn acre$^{-1}$ can prevent Zn deficiency for up to 5 years following application for corn production in New York [69].

**Table 4.** Extractable Cu, Zn, and B in post-harvest soils (0–15 cm) from wheat, pea, and canola plots, respectively.

| Treatment | Cu (mg kg$^{-1}$) | | Zn (mg kg$^{-1}$) | | B (mg kg$^{-1}$) | |
|---|---|---|---|---|---|---|
| | Haverhill | Echo | Haverhill | Echo | Haverhill | Echo |
| C | 0.61 b | 0.65 b | 0.71 b | 1.30c | 0.98 a | 0.94 b |
| B | 1.69 b | 2.27 ab | 2.34 a | 2.12 b | 1.59 a | 1.97 a |
| BI | 1.66 b | 1.27 ab | 1.60 ab | 2.99 a | 1.62 a | 1.90 ab |
| SR | 3.18 a | 2.53 a | 1.90 ab | 1.52 c | 2.15 a | 1.98 a |
| F | 0.63 b | 0.70 b | 0.80 b | 1.36 c | 1.14 a | 1.15 ab |
| *p*-values | <0.0001 | 0.008 | 0.011 | <0.0001 | 0.055 | 0.012 |
| SEM | 0.293 | 0.383 | 0.323 | 0.108 | 0.268 | 0.231 |

Cu, Zn, and B fertilizer were applied for growing wheat, pea, and canola, respectively. Crops were grown at two sites in the field representing two different landscape elevation positions classified as Haverhill (upslope) and Echo (downslope) soil associations, respectively. Treatment evaluation includes C: control; B: broadcast; BI: broadcast and incorporation; SR: seed row banding; and F: foliar methods of fertilizer application. Means followed by the same letter in a column for treatments at the same site are not statistically significant (*p* > 0.05). SEM = Standard error of mean (*n* = 4).

Residual hot water soluble B significantly increased with application of boric acid on Echo soil (Table 4). Similar residual benefits of increasing B concentration in soils

was reported with broadcast and incorporation of B fertilizer [22]. Additionally, soil application of 1 kg B ha$^{-1}$ for rice production did show residual benefits in improving subsequent vegetable yields in India [70]. Overall, soil applied micronutrient fertilizer may be effective in both correcting deficiency in year of application and also contribute to enhanced micronutrient nutrition for following crops in rotation. However, establishment of likelihood of deficiency and crop response to fertilization is needed for both the crop grown that year and crops that follow in order to determine if micronutrient addition will be economical.

Micronutrient ions such as $Cu^{2+}$ and $Zn^{2+}$ tend to be adsorbed strongly through interaction with negatively charge sites on organic matter and clay minerals, and thus can show restricted mobility and availability in soils high in these soil constituents [2,65]. Further, the ion supply rate is adversely affected by dry soil conditions due to increased path length for diffusion as related to increased tortuosity [28,71]. In this study, the supply rates were greatly influenced by fertilizer placement strategies along with the effect of local environments as revealed by PRS™-probes placed in the field plots during the growing season. The cumulative nutrient supply rate measured by in situ burials of PRS™-probes indicated that the Cu and Zn supply in Haverhill soil were significantly higher with broadcast application of corresponding micronutrient fertilizer (Figure 2). Limited rainfall received at this site likely resulted in the applied micronutrient fertilizer remaining in the soil surface horizon, especially in the upper slope Haverhill soil, where it could not be accessed by roots growing deeper in search of moisture but instead was supplied to the PRS$^{TM}$ probe just below the soil surface. Typically, the supply and adsorption of nutrient ion to PRS™-probe membrane was significantly influenced by soil moisture [72]. Insufficient moisture may also cause incomplete contact between the PRS™-probe membrane and the soil, therefore high variability in ion supply rates may occur in the field [28]. Increased variability in B supply due to dry field conditions has been reported in a field trial conducted in northeast Saskatchewan [73].

*3.4. Micronutrients in Soil Post-Harvest*

3.4.1. Chemical Speciation of Cu, Zn, and B

The fate of soil applied micronutrients depends greatly on the chemical form or species into which they are adsorbed. The chemical speciation of Cu, Zn, and B obtained from sequential chemical extraction of soil samples obtained in the fall after the 2015 crop plot harvest is shown in Table 5. Both with and without fertilization, the soil solution-carbonate-exchangeable fraction of Cu and Zn was small in the surface soil (0–15 cm) of both Haverhill and Echo sites relative to other fractions. The residual fraction was the predominant chemical form, constituting approximately 60%, 76% and 92% of the total Cu, Zn, and B, respectively. The second most dominant species was the organic fraction where micronutrients bind through several mechanisms due to the simultaneous presence of numerous adsorption sites in the soil organic matter [74,75]. Among the micronutrient metals, Cu is specifically adsorbed by organic matter to form stable complexes that alter availability to plants [76]. Increased association of micronutrient with organic materials could be attributed to the higher organic matter content and unweathered nature of prairie soils compared to other regions. Similar to our results, other studies with soils from farm fields of the prairies have shown that much of the Cu and Zn is in recalcitrant, resistant fractions, while the lowest proportion is in solution and-exchangeable fractions [44,77–79]. Another study reported that nearly 97% of the total soil B was in residual or occluded form that is recalcitrant, and not readily accessible by plants [34].

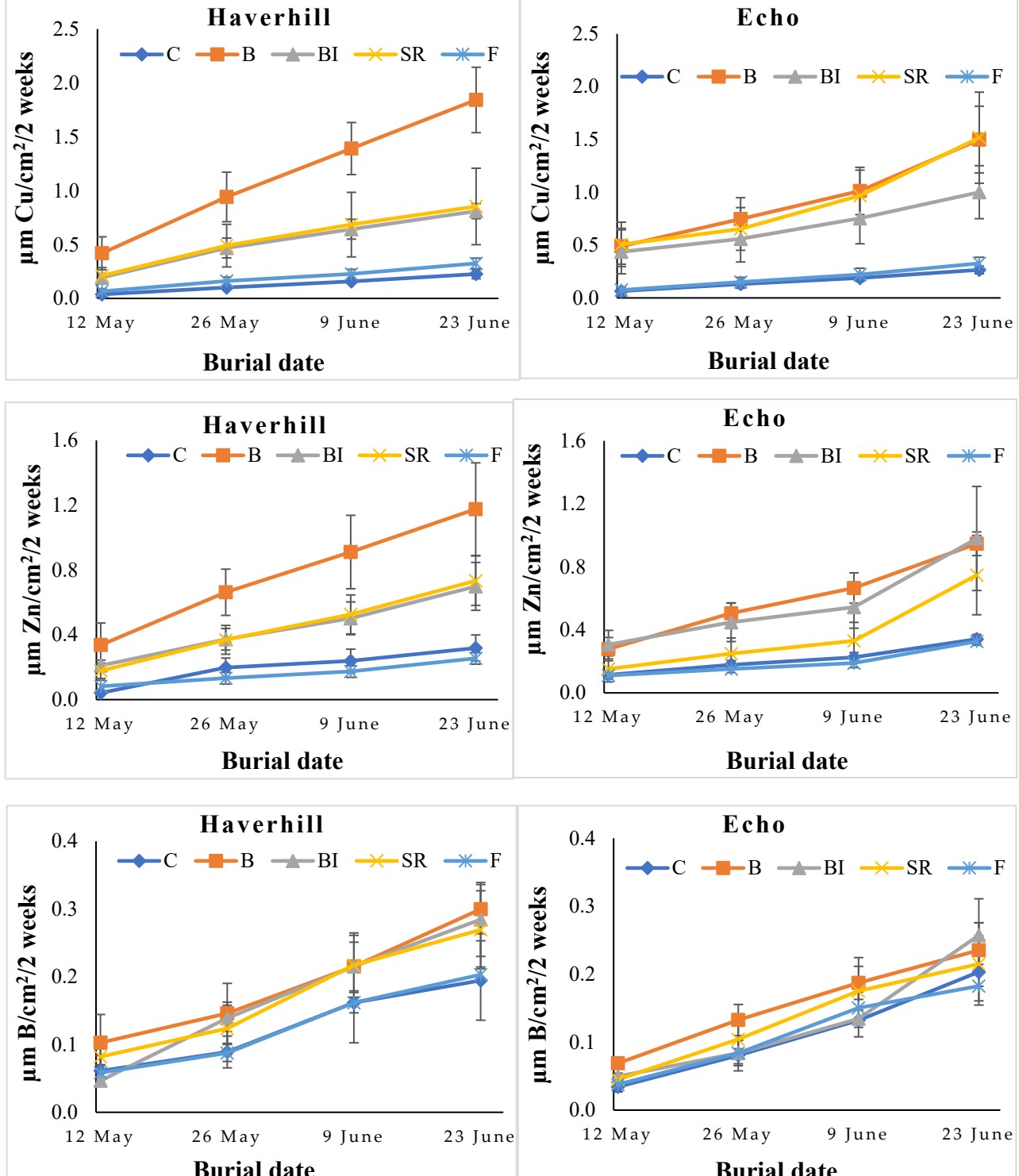

**Figure 2.** Mean cumulative nutrient supply rates of Cu, Zn, and B, measured by in situ burials of Plant Root Simulator (PRS)-probes at two weeks intervals during seeding to maximum vegetative growth stages. The Cu, Zn, and B fertilizer were applied for growing wheat, pea, and canola, respectively. Crops were grown at two sites in the field representing two different landscape elevation positions classified as Haverhill (upslope) and Echo (downslope) soil associations, respectively. Treatment evaluation includes C: control; B: broadcast; BI: broadcast and incorporation; SR: seed row banding; and F: foliar methods of fertilizer application. Error bar represents standard error of mean.

**Table 5.** Sequential extraction of Cu, Zn, and B in post-harvest soil.

| Treatment | Cu (mg kg$^{-1}$) | | | | | Zn (mg kg$^{-1}$) | | | | | B (mg kg$^{-1}$) | | | | |
|---|---|---|---|---|---|---|---|---|---|---|---|---|---|---|---|
| | F$_1$ | F$_2$ | F$_3$ | F$_4$ | F$_5$ | F$_1$ | F$_2$ | F$_3$ | F$_4$ | F$_5$ | F$_1$ | F$_2$ | F$_3$ | F$_4$ | F$_5$ |
| | | | | | | | Haverhill | | | | | | | | |
| C | 0.38 b | 0.53 b | 2.60 a | 4.60 a | 8.11 b | 0.51 a | 3.79 a | 7.56 a | 37.1 a | 48.9 a | 1.32 a | 0.93 a | 2.51 a | 75.3 | 81.0 |
| B | 1.11 a | 0.95 b | 2.58 a | 5.59 a | 10.2 ab | 1.12 a | 5.54 a | 7.82 a | 37.4 a | 51.9 a | 1.42 a | 0.98 a | 2.60 a | 74.8 | 81.4 |
| BI | 1.10 a | 1.06 ab | 3.06 a | 5.77 a | 11.0 a | 1.08 a | 4.94 a | 7.74 a | 37.5 a | 51.3 a | 1.57 a | 1.08 a | 2.70 a | 75.8 | 82.8 |
| SR | 1.18 a | 1.55 a | 3.19 a | 5.91 a | 11.8 a | 1.21 a | 5.20 a | 8.04 a | 37.5 a | 51.9 a | 1.36 a | 1.10 a | 2.57 a | 75.5 | 82.6 |
| F | 0.39 b | 0.63 b | 2.38 a | 4.91 a | 8.31 b | 0.54 a | 3.76 a | 7.80 a | 36.6 a | 48.7 a | 1.42 a | 0.96 a | 2.51 a | 75.0 | 81.0 |
| *p*-values | <0.0001 | 0.0004 | 0.422 | 0.567 | 0.0002 | 0.061 | 0.056 | 0.389 | 0.991 | 0.517 | 0.208 | 0.468 | 0.097 | - | - |
| SEM | 0.044 | 0.129 | 0.391 | 0.657 | 0.493 | 0.200 | 0.427 | 0.182 | 1.60 | 1.76 | 0.897 | 0.098 | 0.081 | - | - |
| | | | | | | | Echo | | | | | | | | |
| C | 0.38 c | 0.94 a | 2.48 a | 5.03 a | 8.82 b | 0.82 b | 3.64 a | 8.16 a | 41.2 a | 53.8 a | 0.65 a | 0.40 a | 2.46 a | 75.4 | 79.8 |
| B | 1.18 ab | 1.39 a | 2.96 a | 5.79 a | 11.3 ab | 1.11 ab | 4.69 b | 7.76 a | 42.8 a | 56.4 a | 0.81 a | 0.46 a | 2.38 a | 76.2 | 81.8 |
| BI | 1.10 b | 1.19 a | 2.58 a | 5.21 a | 10.1 ab | 1.60 a | 5.71 a | 8.40 a | 40.8 a | 56.5 a | 0.83 a | 0.48 a | 2.65 a | 76.2 | 82.1 |
| SR | 1.36 a | 1.41 a | 2.66 a | 6.51 a | 11.9 a | 1.03 b | 3.76 c | 7.95 a | 42.4 a | 55.2 a | 0.73 a | 0.46 a | 2.58 a | 75.6 | 81.4 |
| F | 0.34 c | 0.82 a | 2.16 a | 5.36 a | 8.7 b | 0.79 b | 3.58 c | 7.47 a | 40.7 a | 52.6 a | 0.61 a | 0.38 a | 2.58 a | 75.1 | 79.8 |
| *p*-values | <0.0001 | 0.311 | 0.384 | 0.688 | 0.013 | 0.002 | <0.0001 | 0.131 | 0.727 | 0.248 | 0.106 | 0.18 | 0.097 | - | - |
| SEM | 0.067 | 0.230 | 0.288 | 0.785 | 0.679 | 0.123 | 0.213 | 0.248 | 1.57 | 1.57 | 0.063 | 0.033 | 0.085 | - | - |

Different fractions of micronutrients are F$_1$: soil solution-carbonate-exchangeable fraction (Cu and Zn) or specifically adsorbed fraction (B); F$_2$: oxyhydroxide fraction; F$_3$: organic-bound fraction; F$_4$: residual fraction; F$_5$: total concentration in soil. Cu, Zn, and B fertilizer were applied for growing wheat, pea, and canola, respectively. Crops were grown at two sites in the field representing two different landscape elevation positions classified as Haverhill (upslope) and Echo (downslope) soil associations, respectively. Treatment evaluation includes C: control; B: broadcast; BI: broadcast and incorporation; SR: seed row banding; and F: foliar methods of fertilizer application. Means followed by the same letter in a column for treatments at a site are not statistically significant (*p* > 0.05). SEM = Standard error of mean (*n* = 4).

It is well documented that soil microenvironment and chemical properties, such as pH, redox potential, free lime, and organic matter content largely control the rate of adsorption of micronutrients onto the solid surfaces of soil [75,80]. Most of the Cu and Zn added as fertilizer appear to be retained in the soil solution-carbonate-exchangeable fraction (Table 5), with similar distribution patterns observed among the soil placed treatments at both sites. Both the Haverhill and Echo soils have relatively similar sand and organic carbon contents so similar redistribution among fractions is expected. Typically, the Cu adsorption in soil is greatly influenced by competition between adsorption sites on carbonate minerals and organic matter [81]. Under alkaline pH soil conditions, Cu has a stronger preference to bind with carbonate mineral even in the presence of humic acid [82]. Similarly, it was found that the soil applied Zn fertilizer was mostly distributed to the carbonate-bound fraction in a calcareous field soil from south central Saskatchewan [45]. In general, the bioavailability of soil applied Cu and Zn is often low in soils with pH above 7, similar to that in these study sites. However, the soils are not highly calcareous and, with a surface pH of ~ 7.2 to 7.5, one might expect a combination of carbonate and organic matter associated Cu and Zn. We also found that only small amounts of added Cu and Zn were distributed to oxyhydroxide (F$_2$) fraction, with little evidence for entry into the organic matter (F$_3$) fraction (Table 5). Earlier research [67,83,84] reported that specific adsorption of micronutrient metals to amorphous Fe, Mn and Al oxides is strongly pH dependent and corresponds to the hydrolysis of metal ions. Further, association with the organic matter is a typical stabilization mechanism of micronutrients in soils [76]. Limited entry of Cu or Zn into the organic fractions could be explained by early season dry conditions at the site, and low organic matter content and biological activity in the soil. Apart from hot water soluble B at Echo site, the chemical speciation of B was not significantly influenced by fertilization strategies. Usually, the distribution of B to various chemical forms and adsorption by clay minerals is controlled by soil pH and humic substances [12,85]. Lack of ability to detect effects of fertilization on B distribution in chemically separable fractions may be overcome by a higher application rate than that used in the current study.

### 3.4.2. Spectroscopic Speciation of Cu, Zn, and B

Molecular level understanding of the fixation and transformation processes that soil applied micronutrients undergo can aid in the practical development of best management practices within the farming system, by revealing physicochemical forms and related processes that other methodologies cannot. In this study, bulk X-ray absorption near-edge structure (XANES) spectroscopy revealed that soil Cu was predominantly associated with carbonate, and methoxide phases at both Haverhill and Echo experimental sites. The Cu speciation as revealed by LCF analysis of the XANES data indicated about 20% $CuCO_3$, and 80% methoxide in the unfertilized control soil (Figure 3). Apart from slight redistribution of Cu between these species, no changes in Cu speciation were noticed in response to the soil placement of Cu fertilizer. Conversely, Zn associated with $ZnCO_3$ was found to be the dominant phase in both soils amended with or without Zn-sulfate fertilizer (Figure 4). This agrees with our chemical speciation results which reveal that much of the soil applied Cu and Zn was retained in the solution-carbonate-exchangeable and Fe/Mn oxide bound fractions. The B K-edge XANES spectra were analyzed only qualitatively due to the low B concentrations typically found in agricultural soil samples. However, it did indicate that trigonal species $[B(OH)_3]$ was the dominant phase in these soils.

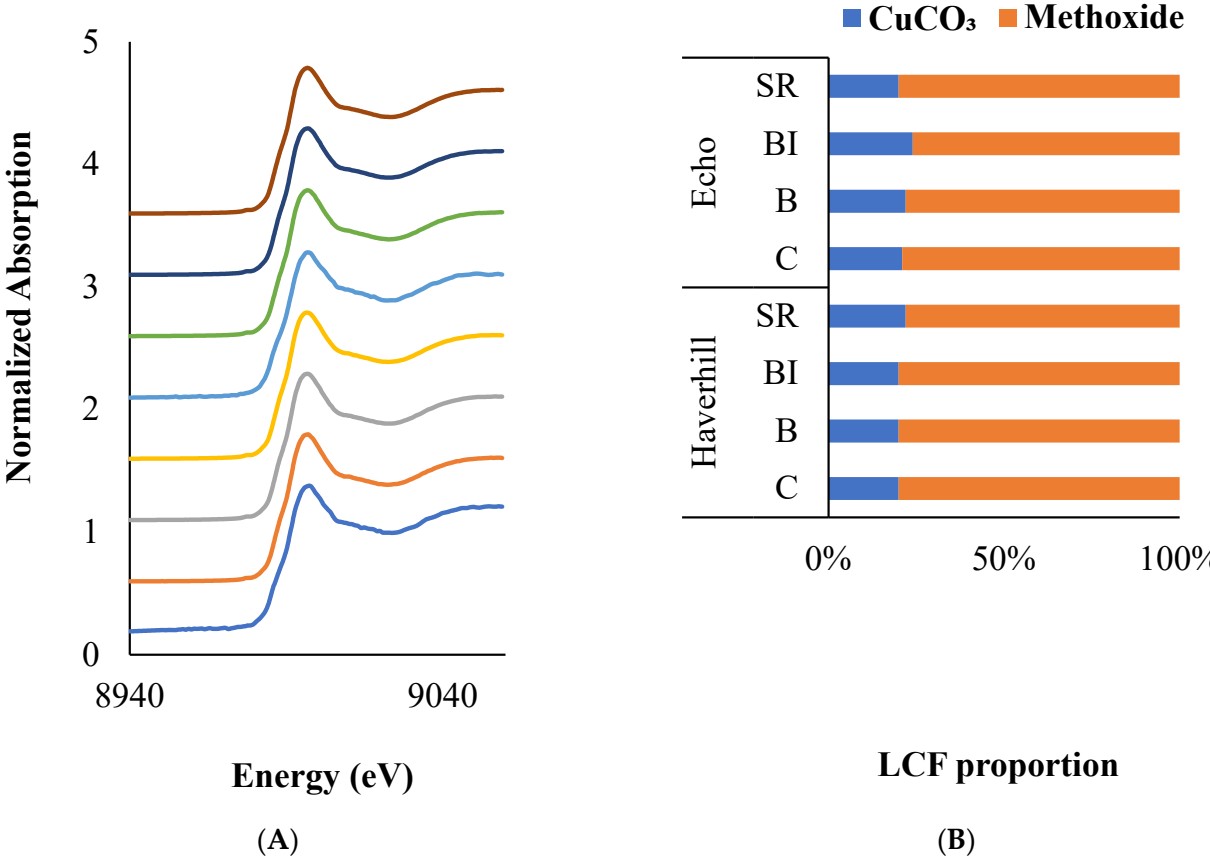

**Figure 3.** (**A**) Normalized Cu XANES K-edge spectra of post-harvest soils collected from landscape-based Cu fertilization field trials in south-central Saskatchewan conducted on Haverhill (upslope) and Echo (downslope) soil associations in a farm field. (**B**) Results of linear combination fit, showing the relative changes in Cu speciation among treatments. Treatments are C: unfertilized control; B: broadcast; Cu BI: broadcast and incorporation Cu; and SR: seed row banding of sulfate form of Cu fertilizer at the rate of 2 kg ha$^{-1}$.

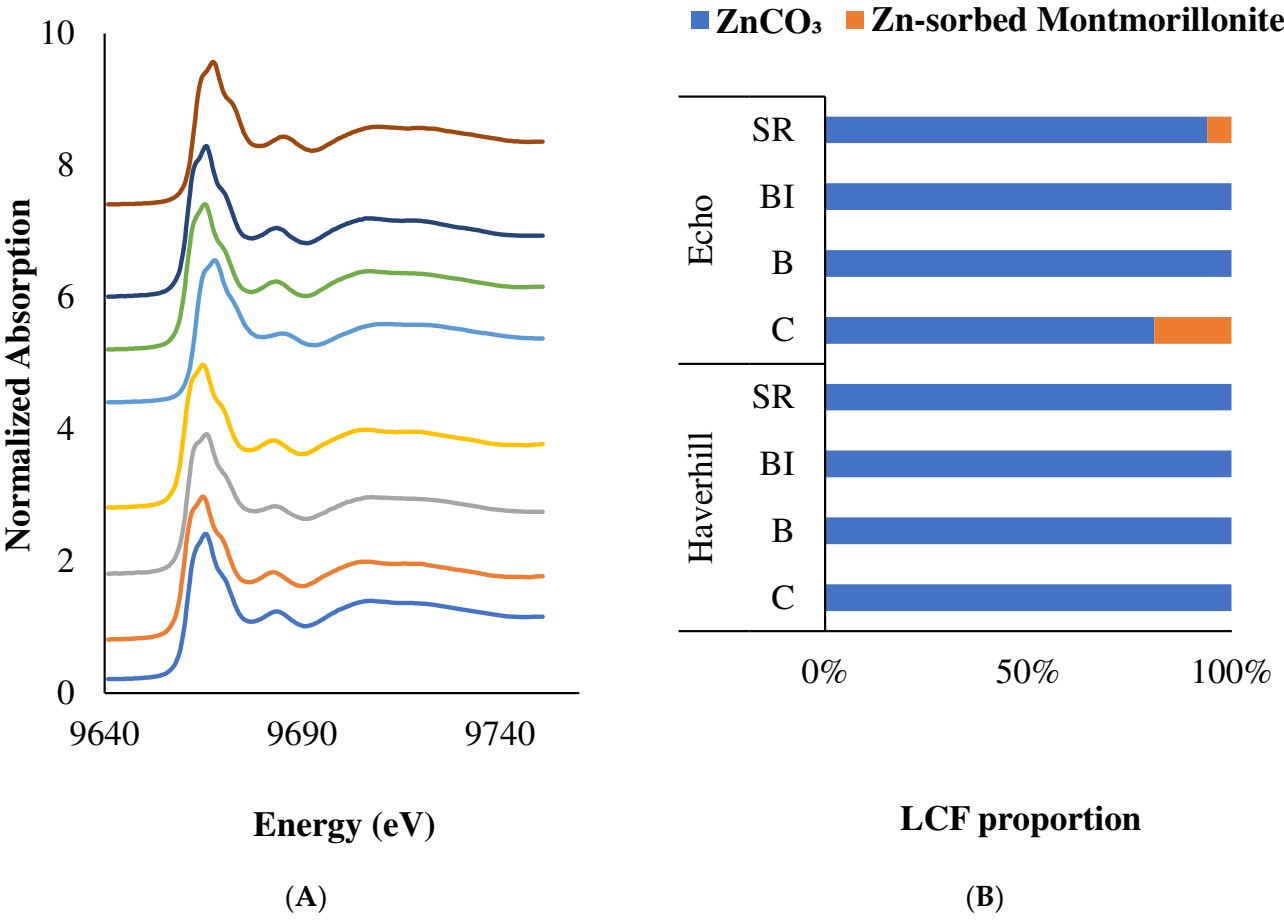

**Figure 4.** (**A**) Normalized Zn XANES K-edge spectra of post-harvest soils collected from landscape-based Zn fertilization field trials in south-central Saskatchewan conducted on Haverhill (upslope) and Echo (downslope) soil associations in a farm field. (**B**) Results of linear combination fit, showing the relative changes in Zn speciation among treatments. Treatments are C: unfertilized control; B: broadcast Zn; BI: broadcast and incorporation Zn; and SR: seed row banding of sulfate form of Zn fertilizer at the rate of 2 kg ha$^{-1}$.

Micronutrient metals such as Cu and Zn have potential for complexation with carbonate minerals, Fe/Mn oxides, and organic components [8,81,86,87]. The bonding nature of Cu and Zn at the calcite surface was previously revealed by EXAFS spectroscopy and showed that both Cu and Zn formed mononuclear inner-sphere adsorption complexes through incorporation of these metals into the Ca site [88]. On the contrary, spectroscopic speciation of Cu in contaminated soils indicated that Cu was preferably speciated into soil organic matter associated forms in comparison to carbonates minerals or Fe/Mn oxides [89]. The presence of similar species in calcareous agricultural soils has also been observed [90,91], which reflects the strong adsorption affinity of Cu to soil organic matter [2,92]. Under natural conditions, sorption of dissolved organic carbon onto soil minerals is a common phenomenon that can further control Cu adsorption on inorganic mineral surfaces [82]. For example, it was found [93] that Cu was strongly complexed with the functional groups of adsorbed organic matter rather than pure alumina surface hydroxyls.

The mobility and bioavailability of Zn in soil is known to be regulated by two main mechanisms: adsorption and coprecipitation. Zinc K-edge EXAFS spectroscopy has identified several Zn species, including octahedrally- and tetrahedrally coordinated adsorbed or complexed Zn, Zn in hydroxy-interlayered minerals (Zn-HIM), Zn-rich phyllosilicates, Zn-layered double hydroxides (Zn-LDH), and hydrozincite in contaminated field soils [94–99]. In addition, it was observed that Zn precipitates (i.e.,Zn-LDH, Zn-phyllosilicate, hydrozincite), and adsorbed/complexed Zn species vary greatly in relation to soil pH and total

Zn content of soils [99]. Usually, Zn precipitation occurs with increased surface loadings under alkaline soil pH due to the saturation of sorption sites [98], whereas the adsorption or inner-sphere complexation is primarily favored under lower loading conditions [100]. Thus, Zn adsorption could be primarily occurring in these soils by ion exchange reaction with montmorillonite, producing adsorbed-Zn or Zn-rich phyllosilicates species.

Similar to Cu and Zn, B can be adsorbed as inner- and outer-sphere complexes on clay minerals surfaces [101,102]. Previous spectroscopic research [12,101–103] reported that both trigonal species [$B(OH)_3$] and tetrahedral species [$B(OH)_4^-$] were coordinated on soil components such as clay minerals, metal oxides, and organic substances. For instance, the B K-edge XANES spectroscopy found the presence of both trigonal and/or tetrahedral inner-sphere complexes on the surface of pure and humic acid coated am-$Al(OH)_3$ [12]. Usually, the trigonal species [$B(OH)_3$] is dominant in soil with a pH range of 5.0 to 7.0, while the tetrahedral species [$B(OH)_4^-$] was found under alkaline soil conditions (pH > 7) [104]. The tetrahedral species [$B(OH)_4^-$] shows a strong affinity to be adsorbed with clay minerals and organic matter, and thus is likely to be less available for plant uptake.

### 4. Conclusions

Different application strategies of Cu, Zn, and B micronutrient did not show significant response to improving wheat, pea and canola yield in two contrasting sites of a farm field in south-central Saskatchewan. However, soil applied micronutrient fertilization resulted in enhanced extractable available micronutrients at harvest from which potential benefits to following crops could be achieved. The chemical and spectroscopic speciation results indicated that much of the soil applied micronutrient that was not taken up by the crop was retained in plant available and potentially labile forms at post-harvest.

**Author Contributions:** N.R. was involved in the experimentation process, performed laboratory analyses, data processing, and manuscript writing. As a supervisor, J.S. provided guideline in project completion, manuscript preparation and editing. All authors have read and agreed to the published version of the manuscript.

**Funding:** This work was financially supported by the Western Grains Research Foundation and Agriculture and Agri-Food Canada Agri- Innovation Program with the project name "Deficiencies in Copper, Zinc and Boron in Prairie Soils", for the period of 2014–2018. The project number is AIP-P034 and the amount is CAD $359,000.

**Institutional Review Board Statement:** Not applicable.

**Informed Consent Statement:** Not applicable.

**Data Availability Statement:** The original data sets of this study are available from the corresponding author on reasonable request.

**Acknowledgments:** We wish to sincerely acknowledge Ning Chen, Weifeng Chen, Lucia Zuin, and Dongniu Wang, for technical assistance at the HXMA and VLS-PGM beamlines during data collection at Canadian Light Source. We also greatly appreciate the valuable contribution of the reviewers regarding the improvement of this manuscript.

**Conflicts of Interest:** The authors declare that they have no conflict of interest.

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
