# Peer review of "Bioavailability, Speciation, and Crop Responses to Copper, Zinc, and Boron Fertilization in South-Central Saskatchewan Soil"

_agronomy, doi:10.3390/agronomy12081837_

Round 1

Reviewer 1 Report

The manuscript was to examine the impact of Cu, Zn, and B fertilizer application strategies on bioavailability and crop responses. The study of trace elements is an ancient topic.There have been many studies on the crop response of trace elements, especially Cu,Zn and B, and the experimental design of this manuscript was not innovative. However, this manuscript has certain significance for accumulating relevant research data.I suggest accepting after major modification.

1. The title need revised. There are no study on mobility of fertilizer.

2.  The abstract  needs to include the main research methods and the main research results data.It is best to put forward the innovation of the manuscript.

3. The whole manuscript needs to be to be condensed.

4. It is necessary to quote relatively new references.

Author Response

Please find attached the point by point reply. Thank you.

Reviewer 2 Report

The authors applied Cu, Zn, and B as fertilizer at two different locations/ soil on wheat, pea, and canola plants and assess the distribution/ translocation, availability in the soil to plants as well as plant productivity/ quality. Overall, the present article is well written and drafted, statistically sound, and the results are fine. The response of micronutrient elements to plants is well-known concept. The current manuscript followed a similar response to earlier reports published. Novelty is missing in this article. Authors include experimental evidence, or plant comparisons (images) must be furnished at different sites/ soils during the demonstration, if possible.

Few minor comments/ suggestions are as stated below:

·         Title should be changed/modified

·         Abstract needs improvement

·         Line 27-44:  Plz improve the language

·       Results and Discussion section need improvement. Some sentences meaning is not clear.

·         Conclusion section is very weak, plz rewrite

·         Please check the typos and grammatical errors in the entire MS.

Author Response

(The authors gave the same response as above.)

Reviewer 3 Report

Overall, the experiment had limited results of the application strategies (4) of Cu, Zn, B on three crops biomass yield, bioavailability and fate of applied microelements. The authors explained the main factors for the limited results, such as adequate nutrition without treatments, low precipitation and carbonate soils. 

Much better and more significant results would be if the experiment was set on Zn, Cu, B deficient soil and if the experiment repeated due to get  more reliable weather conditions.

Even though, as relevant and modern  methods are used in soil analysis, has appropriate discussion, in my opinion the manuscript can be accepted for the publication after minor revision:

- Latin names are usually written italic

- In Table 1, please include CaCO3 (%)

- Table 3, please, change title with: The concentrations of Cu,.....affected by

Author Response

(The authors gave the same response as above.)
